

# Prevalence of oral submucous fibrosis across diverse populations: a systematic review and meta-analysis

Mengqi Wang[*], Chengchen Duan[*], Yuzi Wei and Xiaoping Xu

State Key Laboratory of Oral Diseases & National Center for Stomatology & National Clinical Research Center for Oral Diseases & Research Unit of Oral Carcinogenesis and Management & Chinese Academy of Medical Sciences, West China Hospital of Stomatology, Sichuan University, Chengdu, Sichuan, China
[*] These authors contributed equally to this work.

## ABSTRACT

**Purpose.** The aim is to offer a comprehensive overview of oral submucous fibrosis (OSF) prevalence and explore its epidemiological characteristics across various demographic groups and geographical locations, thereby helping the formulation of public health management policies.

**Methods.** Studies reporting OSF prevalence were identified from electronic databases including the Cochrane Central Register of Controlled Trials (CENTRAL), MEDLINE via PubMed, EMBASE via OVID, and Web of Science. Pooled prevalence and quality assessment using the New-Ottawa Scale were conducted. Two reviewers screened and selected records, assessed quality, and independently extracted data. This systematic review and meta-analysis followed the PRISMA guidelines and was registered on PROSPERO (CRD42024532975).

**Results.** Sixty-three studies, encompassing 11,434 cases in a total population of 769,860, reported OSF prevalence. The majority of studies (forty-one) were from India. The pooled prevalence of OSF across all populations was 3.0% (95% CI [2.8–3.2]%). In non-risk populations, risk populations, and consecutive dental patients, the pooled prevalence stood at 2.4% (95% CI [2.1–2.6]%), 4.5% (95% CI 3.5-5.6%), and 5.6% (95% CI [3.8–7.5]%), respectively. Subgroup analysis by age, sex, and geographical region revealed higher prevalence rates among those aged 50 and above (4.2%, 95% CI [3.0–5.4]%), males (3.3%, 95% CI [2.9–3.7]%), and in India (4.0%, 95% CI [3.7–4.3]%). As the exposure to risk factors exceeded 50% in the surveyed population, the prevalence of OSF notably increased. Most studies met satisfactory quality standards.

**Conclusions.** Our research findings reveal a comprehensive prevalence of OSF across all populations at 3.0% (95% CI [2.8–3.2]%). And, existing evidence indicates a relatively low prevalence of OSF associated with tobacco, alcohol, and Areca (betel) nut use. However, further large-scale studies are recommended to validate these findings. Understanding the prevalence and distribution patterns of OSF may assist in healthcare intervention planning and alleviate the oral cancer burden associated with OSF.

Corresponding author
Xiaoping Xu, xiaoping327@126.com

## INTRODUCTION

Oral submucous fibrosis (OSF) is a chronic disease that progressively restricts mouth opening due to fibrous bands in the oral mucosa (*Vohra et al., 2015*). This condition is commonly linked to betel quid chewing, especially prevalent in South and Southeast Asia. This leads to symptoms such as burning pain and lockjaw, which significantly impair various oral functions, thereby greatly affecting the patient's quality of life (*Bhatt et al., 2019*). OSF is identified by the World Health Organization (WHO) as a high-risk precancerous condition for oral squamous cell carcinoma (OSCC), with reported malignant conversion rates of up to 6% in some studies (*Kujan, Mello & Warnakulasuriya, 2021*; *Warnakulasuriya et al., 2021*). Hence, it poses a considerable threat to long-term health outcomes. Currently, due to its irreversibility, OSF cannot be cured by any effective treatment method, despite ongoing clinical trials and various therapeutic approaches (*Rao et al., 2020*). Prevention remains the sole viable option for clinicians to address OSF.

According to WHO data, there are approximately 5 million cases of OSF worldwide, with distinct geographical patterns (*Yuwanati et al., 2023*). Among countries with a high prevalence of OSF, Southeast Asian nations account for the largest proportion, constituting 10.54% of cases (*Mello et al., 2018*). Predominantly reported in India, OSF cases have also been documented in other regions such as Indonesia, Bangladesh, southern China, Thailand, Sri Lanka, and Nepal (*Yuwanati et al., 2023*). Its prevalence extends beyond Asia, with reported cases in Europe and North America (*Mostafa et al., 2021*). However, the majority of studies reporting the prevalence of OSF are either hospital-based or rely on conveniently selected research cohorts, potentially failing to reflect the true prevalence of OSF. Furthermore, the etiological factors associated with OSF are multifaceted, including autoimmunity (*Gupta et al., 2022*), nutritional deficiencies such as vitamins B and C, and iron (*Wang et al., 2015*), the consumption of spicy foods (*Wang & Tang, 2022*), human papilloma virus (HPV) infection (*Sudhakaran, Hallikeri & Babu, 2019*), genetic mutations (*He et al., 2020*), as well as the consumption of betel nuts, smoking, and alcohol, which are notably correlated with the progression of OSF (*Ray, Chatterjee & Chaudhuri, 2019*). Regions with high rates of betel nut production and consumption often exhibit elevated incidences of OSF (*Rao et al., 2020*). However, not all individuals who indulge in betel nut chewing develop OSF (*Sarode et al., 2013*). Additionally, research indicates that 93% of OSF patients have a history of tobacco product use, with a significant portion also exhibiting alcohol consumption habits (*Karthikeyan, Ramasubramanian & Ramanathan, 2020*). Therefore, the etiology of OSF is currently considered multifactorial and unclear, despite significant research efforts aimed at elucidating its pathogenesis.

To comprehensively delineate the epidemiological profile of OSF, several meta-analyses investigating its prevalence have been conducted. Yuwanati et al. reported a prevalence rate of 5% among betel nut chewers without performing subgroup analyses based on factors such as age, gender, and country (*Yuwanati et al., 2023*). Meanwhile, *Mello et al.*'s *(2018)* meta-analysis indicated a prevalence rate of 4.96% for OSF. However, their exclusion of studies examining population exposure to risk factors may limit the comprehensive representation of OSF prevalence worldwide. This systematic review and

meta-analysis seeks to comprehensively determine the global prevalence of OSF across various demographic groups and geographical locations, thereby filling the existing knowledge gaps in the epidemiology of this condition

## MATERIAL AND METHODS

This systematic review and meta-analysis was conducted based on the Preferred Reporting Items for Systematic Reviews and Meta-Analyses (PRISMA) statement and has been prospectively registered on the PROSPERO website (CRD42024532975) (*Page et al., 2021*).

### Eligibility criteria

The eligibility criteria were formulated in line with the study objectives and background. Studies evaluating or reporting the prevalence of OSF within surveyed populations, or providing the number of OSF patients among the total population surveyed, were considered eligible for inclusion in this systematic review and meta-analysis. There were no restrictions based on age, gender, or publication year. Diagnosis of OSF cases had to be reported as confirmed by professional medical practitioners.

### Search strategy and study selection

A comprehensive search was conducted on the Cochrane Central Register of Controlled Trials (CENTRAL), Medline *via* Ovid, EMBASE, and Web of Science, covering the period from the inception of this research project to Mar 31 2024. The specific search strategies for these databases are detailed in Supplementary Material 1.

The reference lists of the selected studies were manually reviewed to identify any additional relevant studies. The retrieved articles were systematically imported into EndNote X9 (Thomson Reuters, located in Philadelphia, PA, USA) for screening and removal of duplicates. the titles and abstracts of all articles culled from the electronic databases were subjected to a rigorous, independent review by the two researchers (C. D. and M. W). Full-text articles that were considered potentially relevant were subsequently retrieved for an in-depth evaluation. Disagreements were resolved through discussion or the guidance of an arbitrator (X. X.). The reasons for exclusion were recorded for all the full texts.

### Data extraction

Data extraction was conducted independently by two researchers (C. D. and M. W.) utilizing a pre-designed form to systematically compile information after the completion of the search. The following data were extracted: author names, publication dates, study types, countries, data sources, diagnostic criteria, sample characteristics (sample size, age, gender, exposure factors, etc.), and the number of cases diagnosed with OSF. Disagreements were settled through discussion with an arbitrator (X. X.). Prevalence was determined based on the count of OSF diagnoses against the total screened sample size. Additionally, gender, country, and age data were recorded to facilitate subgroup analyses, while data pertaining to exposure factors were documented for subsequent risk correlation assessments. Microsoft Excel was utilized for collecting data.

## Quality assessment

Critical assessment of bias risk was performed using evaluation tools specifically developed for prevalence studies (*Munn et al., 2019*). The Joanna Briggs Institute (JBI) quality evaluation tools for prevalence studies comprise nine items assessing sampling methods, research subjects, and data collection and analysis techniques to evaluate overall study quality. Each item is rated as "yes", "no", "unclear", or "not applicable". Disagreements among reviewers were resolved through discussion with a third expert reviewer.

## Statistical analysis

The data were presented as total sample numbers (N) and OSF cases (n). Pooled prevalence was calculated using double arcsine transformation with the STATA package "meta". Due to the substantial differences in populations and heterogeneous methods of OSF diagnosis across the included studies, a random-effects model was employed to calculate the pooled prevalence rates and their corresponding 95% confidence intervals, thereby establishing the composite prevalence rate. For the same reasons, we did not assess publication bias. Based on the sampling or selection methods used in the included studies to assess oral mucosal status, we categorized the surveyed populations into non-risk populations, risk populations, and consecutive dental patients, and summarized prevalence accordingly. Additionally, subgroup analyses were conducted based on gender, region, and mean age of population with OSF. Moreover, subgroup analyses were also performed according to the percentages of tobacco use, alcohol use, and Areca (betel) nut use reported in the included studies' populations. This was done to investigate the impact of different types and degrees of risk factors for OSF on its prevalence. Statistical heterogeneity within this meta-analysis was assessed using Cochran's Q test in conjunction with the Higgins I-squared statistic ($I^2$).

All data analysis was conducted using R version 4.3.2 and STATA 16.0 software (StataCorp LLC, College Station, TX).

# RESULTS

## Studies selection

A total of 63 studies met the inclusion criteria, with data extracted from diverse populations across various continents, representing a comprehensive analysis of OSF prevalence globally (*Agrawal et al., 2021*; *Al-Attas et al., 2014*; *Ali et al., 2017*; *Alvarez Gomez et al., 2008*; *Amarasinghe et al., 2010*; *Ariyawardana et al., 2007*; *Arjun et al., 2014*; *Bhatnagar et al., 2013*; *Byakodi et al., 2011*; *Chandra & Govindraju, 2012*; *Chatterjee, Gupta & Bose, 2015*; *Chiang et al., 2014*; *Chung et al., 2005*; *Dagli et al., 2008*; *Gupta et al., 1989*; *Gupta et al., 1998*; *Gupta et al., 2012*; *Hallikeri et al., 2018*; *Ikeda et al., 1995*; *Jacob et al., 2022*; *Jose et al., 2023*; *Kaur, Chauhan & Shivakumar, 2023*; *Krishna Priya, Srinivas & Devaki, 2018*; *Kumar et al., 2015*; *Kumar et al., 2019*; *Kumar Srivastava, 2014*; *Lee et al., 2011*; *Liu et al., 2022*; *Mathew et al., 2008*; *Mehrotra et al., 2017*; *Mehrotra et al., 2008*; *Mehrotra et al., 2010*; *Nair et al., 2014*; *Nigam et al., 2014*; *Oakley, Demaine & Warnakulasuriya, 2005*; *Oswal et al., 2023*; *Patil, Bathi & Chaudhari, 2013*; *Patil, Doni & Maheshwari, 2015*; *Pindborg et al., 1968*; *Pindborg et al., 1984*; *Rajendran et al., 1992*; *Rani et al., 2023*; *Reddy et al., 2015*;

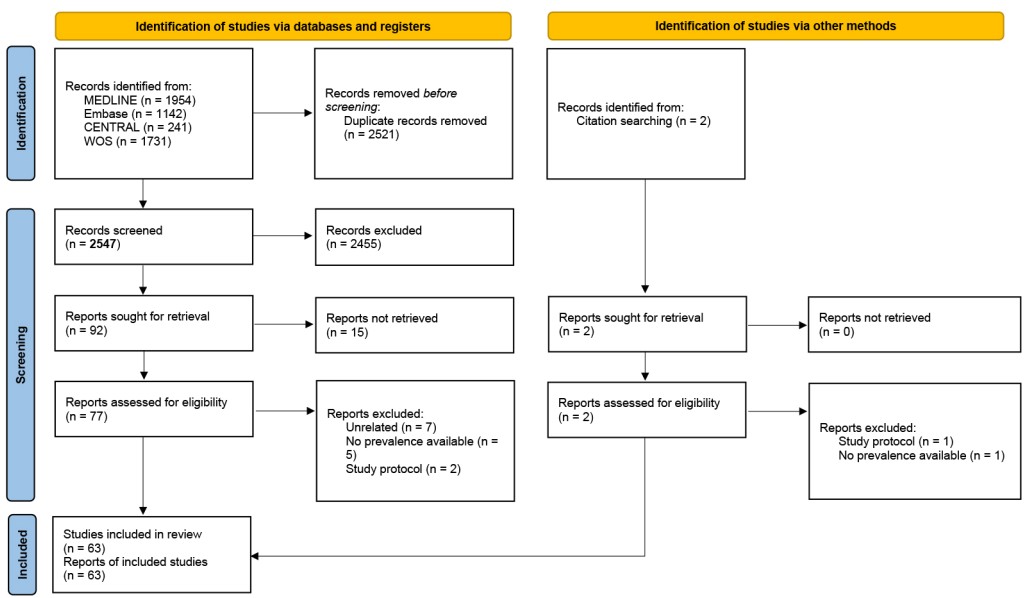

**Figure 1** PRISMA 2020 flow diagram for new systematic reviews which included searches of databases, registers and other sources.

*Reddy et al., 2012*; *Rooban et al., 2009*; *Sankaranarayanan et al., 2000*; *Saraswathi et al., 2006*; *Sari et al., 2024*; *Seedat & Van Wyk, 1988*; *Shrestha et al., 2023*; *Singh et al., 2023*; *Sujatha, Hebbar & Pai, 2012*; *Sumithrarachchi et al., 2024*; *Tang et al., 1997*; *Thavarajah et al., 2006*; *Verma & Sharma, 2019*; *Villa & Gohel, 2014*; *Yang et al., 2005*; *Yang et al., 2010*; *Yang et al., 2001*; *Yunus et al., 2019*; *Zain et al., 1997*; *Zhang et al., 2012a*). The methodology and progression of our research are visually depicted in Fig. 1.

## Study characteristics

The essential characteristics of the studies included in the research are detailed in Table 1. These studies were published between the years 1968 and 2024, covering a total of 11 regions, namely India, China, South Africa, Cambodia, Malaysia, Micronesia, Sri Lanka, the United States, Indonesia, Saudi Arabia, and Nepal. The predominant types of research conducted were epidemiological surveys and cross-sectional studies. Out of the collected articles, 47 reported the time frame for data collection, while 16 did not specify this information. The reported data collection spanned from the year 1900 to 2022.

The studies included a diverse range of surveyed populations, broadly categorized as follows: the general population without specific restrictions, populations with a history of substance use, individuals exposed to radiation, patients seeking medical consultation, and individuals from various occupational backgrounds. Consequently, in synthesizing the data, we classified the studied populations into three groups: non-risk population, risk population, and consecutive dental patients. The non-risk population primarily comprised the general population without specific restrictions based on community settings. The risk population mainly consisted of populations with a history of specific

**Table 1  The characteristics of the studies included in the meta-analysis are outlined.**

| Year | Author | Type of study | Region | Data collection time | Population | Diagnosis |
|---|---|---|---|---|---|---|
| 1968 | Pindborg JJ | An epidemiological survey | India | Not reported | No special restrictions | Clinical grounds |
| 1984 | Pindborg JJ | An epidemiological survey | China | 01/01/1900 to 01/01/1900 | Areca (betel) nut chewers | Clinical and histologic changes |
| 1988 | Seedat HA | An epidemiological survey | South Africa | 01/01/1981 to 12/31/1983 | No special restrictions | Palpable fibrous bands |
| 1989 | Gupta. PC | An epidemiological survey | India | 01/01/1977 to 12/31/1978 | Tobacco users | Standardized diagnostic criteria |
| 1992 | Rajendran R | An epidemiological survey | India | Not reported | High background radiation | Clinical criteria and optional biopsy |
| 1992 | Ikeda N | An epidemiological survey | Cambodian | 07/04/1991 to 07/31/1991 | No special restrictions | Pertinent WHO criteria |
| 1997 | Tang JG | An epidemiological survey | China | Not reported | No Special restrictions | The presence of Palpable fibrous bands |
| 1998 | Zain RB | An epidemiological survey | Malaysia | 01/01/1993 to 12/31/1994 | No special restrictions | Pertinent WHO criteria |
| 1998 | Gupta PC | An epidemiological survey | India | Not reported | No special restrictions | Presence of Palpable fibrous bands |
| 2000 | Sankaranarayanan R | Controlled intervention trial | India | Not reported | No special restrictions | Oral biopsy |
| 2001 | Yang YH | An epidemiological survey | China | 03/01/1997 to 06/30/1997 | No special restrictions | Pertinent WHO criteria |
| 2005 | Chung CH | A cross-sectional community survey | China | 09/01/1998 to 04/30/1999 | No special restrictions | The recommendations of the WHO |
| 2005 | Oakley E | A cross-sectional study | Micronesia | 01/01/2004 to 12/31/2004 | High-school students | WHO criteria |
| 2005 | Yang YH | 6-year follow-up study | China | 01/01/1997 to 12/31/2003 | Those without any lesions seen in 1997 | WHO criteria |
| 2005 | Saraswathi TR | A hospital based cross-sectional study | India | 01/01/2004 to 03/31/2004 | Consecutive patients attended the outpatient department | WHO criteria |
| 2006 | Thavarajah R | A hospital based cross-sectional study | India | 06/01/2002 to 05/31/2003 | Consecutive first-visit patients | Not reported |
| 2007 | Ariyawardana A | A cross-sectional study | Sri Lanka | 02/01/1999 to 09/30/1999 | Labourers employed in tea estate plantations in Sri Lanka | Not reported |
| 2008 | Alvarez Gómez GJ | A descriptive study | American | 01/01/2004 to 06/30/2004 | No special restrictions | Clinical and histological changes |

**Table 1** (*continued*)

| Year | Author | Type of study | Region | Data collection time | Population | Diagnosis |
|------|--------|---------------|--------|---------------------|------------|-----------|
| 2008 | Dagli RJ | A cross-sectional study | India | 02/01/2007 to 02/28/2007 | "Green Marble Mines" laborer | WHO criteria |
| 2008 | Mathew AL | A cross-sectional study | India | 03/01/2005 to 06/01/2005 | Consecutive first-visit patients | WHO criteria and biopsy |
| 2008 | Mehrotra R | a single institutional retrospective study | India | 01/01/1990 to 12/31/2007 | Patients with biopsy records | Biopsy records |
| 2009 | Rooban T | A cross-sectional study | India | 06/01/2002 to 05/31/2004 | Alcohol misusers | WHO criteria |
| 2010 | Amarasinghe HK | A cross-sectional community-based study | Sri Lanka | 11/01/2006 to 11/30/2007 | No special restrictions | The recommendations of WHO |
| 2010 | Mehrotra R | A cross-sectional study | India | 05/01/2008 to 06/30/2008 | No special restrictions | The recommendations of WHO |
| 2010 | Yang YH | A cross-sectional community-based study | China | 10/01/2005 to 12/31/2005 | No special restrictions | The recommendations of WHO |
| 2011 | Byakodi R | A cross-sectional study | India | 01/01/2009 to 06/30/2010 | No special restrictions | WHO criteria and biopsy |
| 2012 | Chandra P | A cross-sectional study | India | Not reported | Consecutive first-visit patients | The recommendations of WHO |
| 2011 | Lee, C | A cross-sectional study | China | 01/01/2009 to 02/28/2010 | No special restrictions | The recommendations of WHO |
| 2011 | Lee, C | A cross-sectional study | China | 01/01/2009 to 02/28/2010 | No special restrictions | The recommendations of WHO |
| 2011 | Lee, C | A cross-sectional study | Indonesia | 01/01/2009 to 02/28/2010 | No special restrictions | The recommendations of WHO |
| 2012 | Gupta T | A cross-sectional study | India | Not reported | Illicit drug users | The recommendations of WHO |
| 2012 | Reddy V | A cross-sectional study | India | 10/01/2009 to 03/31/2010 | Prison population for Karnataka | The recommendations of WHO |
| 2012 | Sujatha D | A cross-sectional study | India | 07/01/2010 to 03/31/2011 | Areca (betel) nut chewers or alcohol misusers | Presence of blanching and palpable fibrous bands |
| 2012 | Zhang SS | A cross-sectional study | China | 09/01/2009 to 11/30/2009 | No special restrictions | Presence of palpable fibrous bands or palpable stiffness of a large area and blanching of the mucosa of a large area |
| 2013 | Bhatnagar P | A cross-sectional study | India | 05/01/2010 to 10/31/2010 | No special restrictions | Not reported |
| 2013 | Patil PB | A hospital-based, cross-sectional study | India | Not reported | No special restrictions | WHO criteria |

**Table 1** (*continued*)

| Year | Author | Type of study | Region | Data collection time | Population | Diagnosis |
|------|--------|---------------|--------|---------------------|------------|-----------|
| 2014 | Al-Attas SA | A cross-sectional study | Saudi Arabia | 01/01/2011 to 12/31/2013 | No special restrictions | Not reported |
| 2014 | Arjun TN | A cross-sectional survey | India | 09/01/2013 to 09/14/2013 | Psychiatric inmates and nonpsychiatric inmates residing in central jail | WHO criteria |
| 2014 | Chiang M | A cross-sectional survey | China | 01/01/2003 to 12/31/2007 | Consecutive dental patients | WHO guidelines |
| 2014 | Kumar Srivastava V | case control study | India | Not reported | No special restrictions | Not reported |
| 2014 | Nair PP | A cross-sectional survey | India | Not reported | No special restrictions | Not reported |
| 2014 | Nigam NK | A cross-sectional survey | India | Not reported | Habitual chewers | Not reported |
| 2014 | Villa A | A cross-sectional survey | American | 07/01/2013 to 02/28/2014 | No special restrictions | Not reported |
| 2015 | Chatterjee R | A cross-sectional survey | India | Not reported | Areca-nut Chewing Population | The presence of palpable fibrous bands in the oral mucosa leading to limited mouth opening |
| 2015 | Kumar S | A cross-sectional survey | India | 03/01/2014 to 06/30/2014 | No special restrictions | Not reported |
| 2015 | Patil S | A cross-sectional survey | India | 09/01/2008 to 10/31/2012 | 60 to 98 years, attending the Department of Oral Medicine and Radiology | WHO guidelines |
| 2015 | Reddy SS | A cross-sectional survey | India | 01/01/2010 and 01/01/2012 | Individuals above 18 years using any type of CT for more than 6 months | International recommendations |
| 2017 | Ali AK | A cross-sectional survey | India | 03/01/2016 to 07/31/2016 | Construction workers | WHO guidelines |
| 2017 | Mehrotra D | A cross-sectional survey | India | Not reported | No special restrictions | Not reported |
| 2018 | Hallikeri K | hospital-based cross-sectional study | India | 01/01/2012 to 12/31/2013 | Having history of chewing form of tobacco-related habit, areca nut, and/or betel quid for a minimum 5 years | Clinical examination and biopsy |
| 2018 | Krishna Priya M | A cross-sectional survey | India | 09/06/2011 to 11/28/2011 | With the habit of tobacco and alcohol consumption in various forms | WHO guidelines |

**Table 1** (*continued*)

| Year | Author | Type of study | Region | Data collection time | Population | Diagnosis |
|---|---|---|---|---|---|---|
| 2019 | Kumar S | A cross-sectional survey | India | 07/01/2017 to 06/30/2018 | Patients attending the outpatient department | WHO guidelines |
| 2019 | Verma S | A descriptive, cross-sectional study | India | 07/01/2016 to 09/30/2016 | Participants who visited the Oral Medicine and Radiology Department of Rungta College of Dental Science and Research | Type III clinical examination |
| 2019 | Yunus G Y | A descriptive, cross-sectional study | India | 09/01/2017 to 09/30/2017 | Beedi rolling workers | World Health Organization assessment form of oral premalignant lesions and conditions |
| 2021 | Agrawal S | cross-sectional observational study | India | 05/01/2020 to 02/28/2021 | Patients coming in pre-anaesthesia clinic | Mucosal blanching, burning sensation, restricted mouth opening and presence of fibrous bands |
| 2022 | Jacob LB | A community-based, cross-sectional study | India | 01/01/2019 to 06/30/2019 | No special restrictions | Not reported |
| 2022 | Liu Y | A cross-sectional survey | China | 2020-11-01 to 2021-08-31 | Non-doctor/nurse and non-police officer | The presence of palpable fibrous bands or palpable stiffness of a large area and blanching of the mucosa of a large area |
| 2022 | Liu Y | A cross-sectional survey | China | 2020-11-01-2021-08-31 | Doctor/nurse population | The presence of palpable fibrous bands or palpable stiffness of a large area and blanching of the mucosa of a large area |
| 2022 | Liu Y | A cross-sectional survey | China | 2020-11-01-2021-08-31 | Police officer population | The presence of palpable fibrous bands or palpable stiffness of a large area and blanching of the mucosa of a large area |
| 2023 | Jose C | A cross-sectional survey | India | Not reported | Cultivation and processing of areca nut and its products | Not reported |

(*continued on next page*)

**Table 1** (*continued*)

| Year | Author | Type of study | Region | Data collection time | Population | Diagnosis |
|------|--------|---------------|--------|---------------------|------------|-----------|
| 2023 | Kaur A | A cross-sectional survey | India | Not reported | Textile mill workers | WHO criteria |
| 2023 | Oswal K | A cross-sectional survey | India | 09/01/2021 to 11/30/2021 | 35–60 year Women have a past history of consuming any form of tobacco products | WHO criteria |
| 2023 | Rani P | A cross-sectional survey | India | Not reported | Patients visiting the outpatient department | Not reported |
| 2023 | Shrestha G | A cross-sectional survey | Nepal | 05/01/2019 to 12/31/2019 | No special restrictions | Not reported |
| 2023 | Singh G | A cross-sectional survey | India | 03/01/2021 to 08/31/2022 | With a history of tobacco usage | Not reported |
| 2024 | Sari EF | A cross-sectional survey | Indonesia | 09/01/2016 to 01/31/2017 | No special restrictions | Not reported |
| 2024 | Sumithrarachchi S | A cross-sectional survey | Sri Lanka | 01/01/2020 to 09/30/2020 | No special restrictions | Not reported |

substance use, potentially associated with OSF risk factors. Consecutive dental patients primarily included those attending the outpatient department in a consecutive manner. Regarding the diagnostic criteria for OSF, only 46 articles provided information on the basis of diagnosis. Among these, only 29 explicitly stated that they followed the diagnostic guidelines as outlined by the WHO documents. The remaining studies utilized various diagnostic standards, including palpable fibrous bands, palpable stiffness over a large area, blanching of the mucosa over a large area, limited mouth opening, and oral biopsy.

## Risk of bias within studies

Among the 63 included articles, eight were recognized for their exceptional quality, each earning 9 "yes" (Supplementary Material 2). The majority of studies scored between 6 to 8 'yes', suggesting good quality. Nevertheless, nine articles scored 3 'yes' or less. These studies were mostly published earlier and lacked standard diagnostic methods. Additionally, these articles often lacked a thorough explanation regarding critical aspects such as sample size analysis, sampling methodology, and data analysis. In some studies, the data provided was insufficient, and the research did not adhere to standard statistical methodologies for analysis.

## Prevalence of OSF

Our research findings reveal a comprehensive prevalence of OSF across all populations at 3.0% (95% CI = 2.8−3.2%). Furthermore, based on the sampling or selection methods employed in the included studies to assess oral mucosal status, we categorized the surveyed populations into non-risk populations, risk populations, and consecutive dental patients. Our results indicate that the prevalence of OSF is lowest in the non-risk population, standing at 2.4% (95% CI = 2.1−2.6%), higher in the risk population at 4.5% (95%

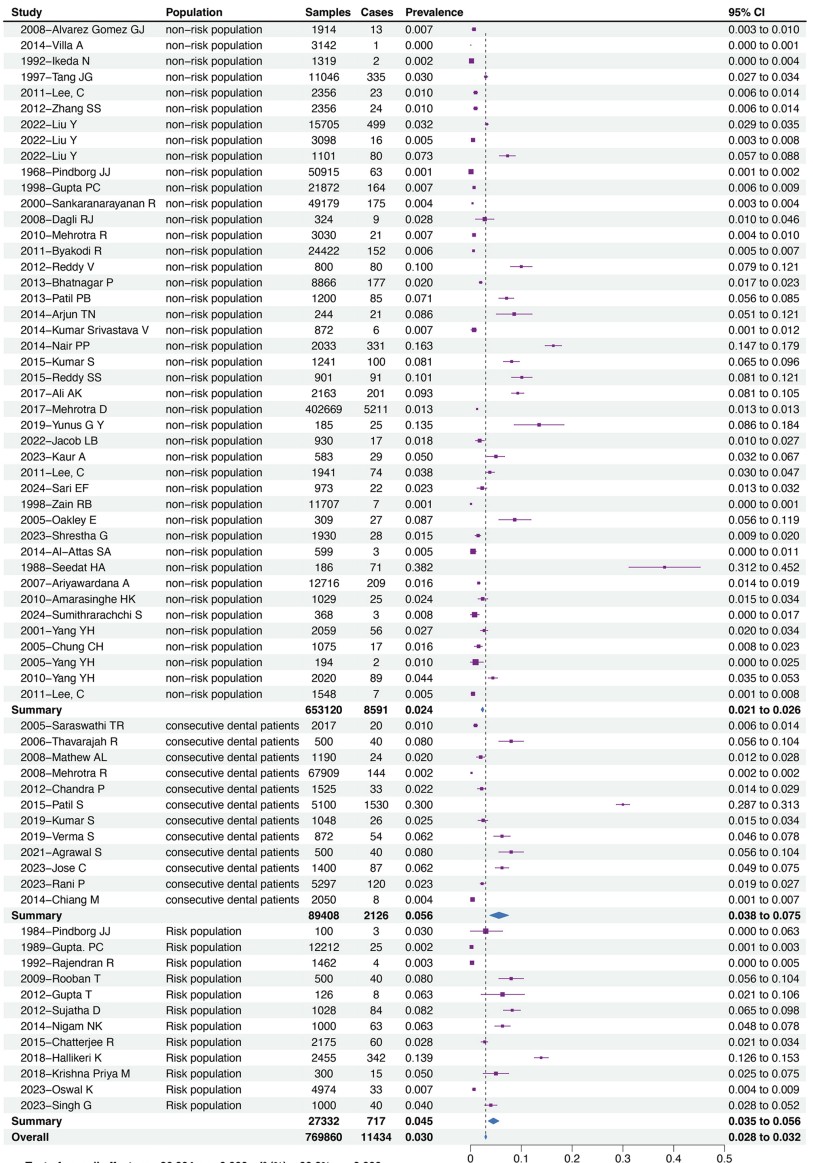

**Figure 2 Forest plot of prevalence of OSF among different population.** This is a forest plot illustrating the prevalence of OSF in different populations and the overall prevalence. It categorizes the surveyed groups based on the literature included, with the following order from top to bottom: non-risk population, consecutive dental patients, and risk population. The chart lists the original data and the calculated prevalence rates from each study, and then summarizes the OSF prevalence in these groups. As depicted in the chart, the prevalence of OSF in the non-risk population is 2.4% (95% CI = 2.1−2.6%); in consecutive dental patients, it is 5.6% (95% CI = 3.8−7.5%); and in the risk population, it is 4.5% (95% CI = 3.5−5.6%). The overall prevalence of OSF across all populations is 3.0% (95% CI = 2.8−3.2%).

CI = 3.5−5.6%), and highest in consecutive dental patients, reaching 5.6% (95% CI = 3.8−7.5%). Figure 2 illustrates the comprehensive prevalence of OSF across all populations and the respective prevalence of OSF within each population.

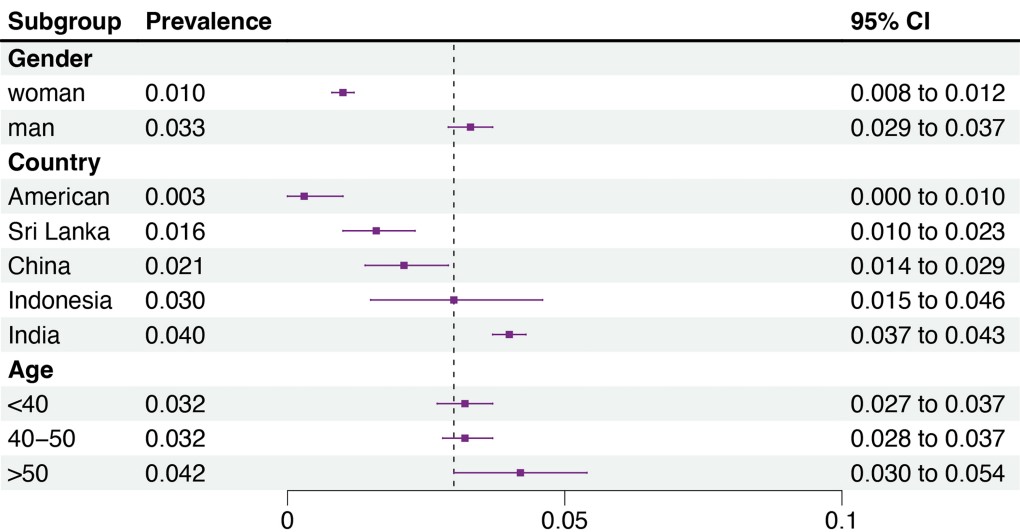

| Subgroup | Prevalence | | 95% CI |
|---|---|---|---|
| **Gender** | | | |
| woman | 0.010 | | 0.008 to 0.012 |
| man | 0.033 | | 0.029 to 0.037 |
| **Country** | | | |
| American | 0.003 | | 0.000 to 0.010 |
| Sri Lanka | 0.016 | | 0.010 to 0.023 |
| China | 0.021 | | 0.014 to 0.029 |
| Indonesia | 0.030 | | 0.015 to 0.046 |
| India | 0.040 | | 0.037 to 0.043 |
| **Age** | | | |
| <40 | 0.032 | | 0.027 to 0.037 |
| 40–50 | 0.032 | | 0.028 to 0.037 |
| >50 | 0.042 | | 0.030 to 0.054 |

**Figure 3  Subgroup analysis of the prevalence of OSF based on gender, age, and country.** This is a forest plot for the subgroup analysis of OSF prevalence by gender, country, and age. By collecting data on the gender, country, and age of patients from the included literature, it is observed that in the gender subgroup analysis, males have a higher prevalence of OSF at 3.3%, while females have a prevalence of 1.0%. In the country subgroup analysis, India has the highest OSF prevalence at 4.0%, followed by Indonesia at 3%, China at 2.1%, Sri Lanka at 1.6%, and the United States shows the lowest prevalence at 0.3%. In the age subgroup analysis, individuals over 50 years old exhibit the highest OSF prevalence at 4.2%, whereas the prevalence is relatively lower, around 3.2%, in individuals under 40 years old and those aged 40–50 years.

Additionally, subgroup analysis of OSF prevalence based on gender, age, and region is presented in Fig. 3. Notably, we observed a higher prevalence of OSF in males at 3.3% (95% CI = 2.9−3.7%) compared to females at 1.0% (95% CI = 0.8−1.2%). Regionally, India exhibits the highest prevalence of OSF at 4.0% (95% CI = 3.7−4.3%), while the United States demonstrates the lowest prevalence at 0.3% (95% CI = 0.0−1.0%). Regarding age, individuals over 50 years old exhibit the highest OSF prevalence at 4.2% (95% CI = 3.0−5.4%), whereas the prevalence is relatively lower around 3.2% in individuals under 40 years old and those aged 40–50 years.

The subgroup analysis results for different types and degrees of risk factors contributing to OSF are depicted in Fig. 4. The pooled prevalence of OSF was found to be higher in populations with a predominant habit of betel quid chewing, as compared to other regions where this habit is less common, indicating a strong correlation between the two factors. Meta-regression analysis further suggests that tobacco use predominantly explains the heterogeneity source (see Supplementary Materials 3−5).

## DISCUSSION

The present meta-analysis revealed that the prevalence of OSF was 3.0% in the overall population, 2.4% in the non-risk population, and reached 5.6% in consecutive dental patients. Despite the comprehensive nature of the data considering factors such as gender, age, habits, region, and exposure to risk factors, it should be noted that the included

| Subgroup | Prevalence | | 95% CI |
|---|---|---|---|
| **Alcohol use** | | | |
| 0–0.3 | 0.024 | | 0.019 to 0.028 |
| 0.3–0.5 | 0.015 | | 0.008 to 0.022 |
| 0.5–1 | 0.058 | | 0.007 to 0.109 |
| **Areca (betel) nut use** | | | |
| 0–0.3 | 0.030 | | 0.021 to 0.038 |
| 0.3–0.5 | 0.020 | | 0.015 to 0.025 |
| 0.5–1 | 0.056 | | 0.041 to 0.071 |
| **Tobacco use** | | | |
| 0–0.3 | 0.017 | | 0.014 to 0.021 |
| 0.3–0.5 | 0.034 | | 0.028 to 0.040 |
| 0.5–1 | 0.072 | | 0.048 to 0.096 |

**Figure 4 Subgroup analysis of the prevalence of OSF based on tobacco use, alcohol use, and betel nut consumption among the populations included.** Subgroup analysis of OSF prevalence under different types and degrees of risk factors: When the percentage of alcohol use, areca (betel) nut use, and tobacco use in the surveyed population exceeds 50%, the prevalence of OSF significantly increases, to 5.8% (95% CI = 0.7–10.9%), 5.6% (95% CI = 4.1−7.1%), and 7.2% (95% CI = 4.8−9.6%), respectively. Moreover, only tobacco use shows a phenomenon where the incidence of OSF rises with the increase in the percentage of tobacco use in the surveyed population. Meta-regression analysis further suggests that tobacco use predominantly explains the heterogeneity source (see Supplementary Materials 3−5).

studies were all cross-sectional, which introduces heterogeneity and limitations (such as high risk of bias) to the original data. Therefore, these estimates should be interpreted with caution. Nevertheless, given the irreversible nature of OSF as a precancerous lesion and its significant economic impact on individuals and healthcare systems, these figures remain concerning and warrant further attention.

This review examined the prevalence of OSF across different genders, countries, and age groups. The results indicate a threefold higher prevalence of OSF in males compared to females. Disparities in the male-to-female ratio of OSF patients were observed across various regions, seemingly correlated with chewing habits prevalent in those areas (*Hazarey et al., 2007*; *Shih et al., 2019*; *Yang et al., 2018*). India reported the highest OSF prevalence, while the United States reported the lowest. Consistent with previous studies, the Indian subcontinent exhibits a high prevalence of OSF (*More et al., 2018*), with India having the highest incidence of OSF among the studied nations (*Ray, Chatterjee & Chaudhuri, 2019*). The elevated prevalence in India is attributed to the increased popularity of commercially prepared Areca betel nut products (*Cai & Huang, 2022*). Developed countries have consistently shown lower OSF incidence rates, possibly due to the presence of substantial migrant populations (*Shen et al., 2020*). Furthermore, the higher prevalence observed in socioeconomically disadvantaged areas is a cause for concern, as these regions often indicate relatively poorer access to healthcare services and lower levels of treatment

adherence, factors that can have a significant impact on disease progression. Additionally, while the prevalence of OSF tends to be higher in individuals over 50 compared to younger demographics, the difference is not significant.

The pathogenesis of OSF is multifaceted, involving inflammation, hypoxia, and reactive oxygen species generation (*Lin et al., 2019*; *Mohammed et al., 2015*; *Shen et al., 2020*). While OSF etiology is multifactorial, extensive research has highlighted habits such as betel nut chewing, tobacco use, and alcohol consumption as significant risk factors for OSF (*Aishwarya et al., 2017*; *Liu et al., 2015*; *Zhang et al., 2012b*). A previous meta-analysis reported a low prevalence (5%) of OSF among areca nut chewers, contradicting the strong association between areca nut chewing and OSF (*Yuwanati et al., 2023*). Our meta-analysis conducted subgroup analyses based on the percentage of Areca betel nut use, tobacco use, and alcohol use within the surveyed populations, revealing a pronounced increase in OSF prevalence when these risk factors were prevalent. This trend is most notable in tobacco use. Consistent with previous findings, concurrent use of betel nut and tobacco significantly escalates OSF incidence, suggesting a synergistic effect of tobacco consumption and betel nut chewing on OSF occurrence (*Cirillo et al., 2022*). Although current epidemiological research underscores betel nut chewing as the foremost risk factor for OSF, this study implies that betel nut may not be the sole causative agent of OSF. Thus, previously underappreciated risk factors for OSF, such as tobacco use, warrant greater attention, prompting the formulation of pertinent public health policies and health promotion initiatives.

This study introduces the systematic secondary review of literature on prevalence of OSF, aiming to estimate the overall incidence by integrating multifactorial, cross-population data. It holds significant value for uncovering the societal health burden of OSF, providing a scientific foundation for the development of public health care plans, and assessing the effectiveness of prevention and treatment strategies. While most existing studies concentrate on Southeast Asia, limiting a comprehensive view of the global prevalence of OSF. This research calls for broader geographical research to fill gaps in our knowledge, as the dynamic progression of OSF and undefined etiology suggest that low-prevalence regions could reveal additional contributing factors beyond diet. Besides, quality assessment revealed methodological deficiencies and inconsistent diagnostic criteria, causing heterogeneity in secondary research. Future research should prioritize methodological clarity and diagnostic standardization to enhance the reliability of findings and support robust public health policy-making in addressing OSF challenges.

While the overall quality of the studies included remains relatively low, this meta-analysis significantly mitigated the adverse effects of heterogeneity on the results by stratifying the data. By categorizing the sampled populations into three groups based on the sampling method, this meta-analysis greatly reduced the impact of sampling methodology-induced heterogeneity, thus making the pooled prevalence of OSF more closely approximate the true value. OSF Additionally, subgroup analyses based on gender, geographical region, and age demonstrated epidemiological characteristics of OSF. Furthermore, subgroup analyses based on variations in the types and extents of risk factor exposures within the surveyed populations provided additional perspectives for preventing and treating OSF. However,

our study still possesses certain limitations. Firstly, not all studies reported their diagnostic criteria for OSF, potentially introducing measurement bias due to variations in the level of experience and subjective perspectives among examining physicians. Secondly, we did not further stratify the risk population into subgroups based on different levels of risk factors exposure, which may have limited the depth of our results. Moreover, despite conducting subgroup analyses, substantial heterogeneity still existed, indicating that we may not have fully elucidated the sources of heterogeneity. Oral manifestations of OSF may not have been evident during the data collection period due to the variability in disease severity at its onset and the significant heterogeneity in treatment response. Consequently, conducting longitudinal investigations becomes crucial in order to provide a deeper understanding of the prevalence of oral symptoms associated with OSF at various stages of the disease.

## CONCLUSIONS

Our research findings reveal a comprehensive prevalence of OSF across all populations at 3.0% (95% CI = 2.8−3.2%). And, existing evidence indicates a relatively low prevalence of OSF associated with tobacco, alcohol, and Areca (betel) nut use. However, further large-scale studies are recommended to validate these findings. Understanding the prevalence and distribution patterns of OSF may assist in healthcare intervention planning and alleviate the oral cancer burden associated with OSF.

## ACKNOWLEDGEMENTS

We would like to express our sincere gratitude to Dr. Chen Qianming for his invaluable guidance and constructive suggestions, which have significantly contributed to the improvement of this manuscript.

### Funding
This research was supported by the Research Funding from West China School/Hospital of Stomatology Sichuan University (No. RCDWJS2022-17 and RD-02-202205), the CAMS Innovation Fund for Medical Sciences (CIFMS, 2019-I2M-5-004), the National Natural Science Foundation of China (No. 82273165), the Sichuan Science and Technology Program (No. 2022NSFSC1495), and Major Project of Natural Science Foundation of Hunan Province (open competition, 2021JC0002). The funders had no role in study design, data collection and analysis, decision to publish, or preparation of the manuscript.

### Grant Disclosures
The following grant information was disclosed by the authors:
The Research Funding from West China School/Hospital of Stomatology Sichuan University: RCDWJS2022-17, RD-02-202205.
The CAMS Innovation Fund for Medical Sciences: CIFMS, 2019-I2M-5-004.
The National Natural Science Foundation of China: 82273165.

The Sichuan Science and Technology Program: 2022NSFSC1495.

Major Project of Natural Science Foundation of Hunan Province: 2021JC0002.

## Competing Interests

The authors declare there are no competing interests.

## Author Contributions

- Mengqi Wang performed the experiments, analyzed the data, prepared figures and/or tables, authored or reviewed drafts of the article, and approved the final draft.
- Chengchen Duan performed the experiments, analyzed the data, prepared figures and/or tables, authored or reviewed drafts of the article, and approved the final draft.
- Yuzi Wei analyzed the data, prepared figures and/or tables, and approved the final draft.
- Xiaoping Xu conceived and designed the experiments, prepared figures and/or tables, and approved the final draft.

## Data Availability

The raw data and code are available in the Supplementary Files.

## Supplemental Information

Supplemental information for this article can be found online at http://dx.doi.org/10.7717/peerj.18385#supplemental-information.

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
