# Peer review of "Prevalence of oral submucous fibrosis across diverse populations: a systematic review and meta-analysis"

_PeerJ, doi:10.7717/peerj.18385_

## Round 0.1 · original submission · Minor Revisions

Dear authors,

Please refer to the reviewers' comments, all unanimously suggesting MINOR revisions. Many thanks for your patience and submission.

Reviewer 1 ·

Basic reporting

1. In introduction, line no 66: authors can add other causative factors of OSF which include autoimmunity, nutritional deficiencies including vitamin B, C, and iron deficiencies, consumption of spicy foods, human papilloma virus (HPV) infection, and genetic mutations.

2. I commend the authors for their comprehensive search of related articles and systematic meta- analysis of the topic and for providing the raw data and supplemental files.

3. In addition, the manuscript is clearly written in professional, unambiguous language.

Experimental design

No comment.

Validity of the findings

All required data for meta-analysis is present.
conclusion is well stated. authors can add the final result i.e. prevalence of OSF, in the first line of conclusion.

Reviewer 2 ·

Basic reporting

No comment

Experimental design

The experimental design does not show the search terms and buleans used for data search across all databases. Yet, many articles can be found on Google serach that are not published in indexed articles so authors shoud justify the exclusion.

Validity of the findings

No comments

Additional comments

The article is comprehensive and depicts importance of OSMF prevalance. However, certain articles have mentioned higher prevalence rate in Indian subcontinent like an article by More CB et al. (2018) and should be included in discussion.

Reviewer 3 ·

Basic reporting

>The language is mostly clear, some sentences, particularly those in the introduction and results sections, could be further simplified for better readability without losing technical accuracy.

1. Introduction section, this statements could be revised: (Oral Submucous Fibrosis (OSF) is a chronic, insidious disease characterized by the progressive inability to open the mouth due to fibrous bands forming within the oral mucosa, often associated with betel quid chewing, which is a significant etiological factor particularly in South and Southeast Asian populations). Suggestion i.e. (Oral Submucous Fibrosis (OSF) is a chronic disease that progressively restricts mouth opening due to fibrous bands in the oral mucosa. This condition is commonly linked to betel quid chewing, especially prevalent in South and Southeast Asia)

2. This systematic review and meta-analysis seeks to comprehensively determine the global prevalence of OSF across various demographic groups and geographical locations, thereby filling the existing knowledge gaps in the epidemiology of this condition. revise it.

3. Results Section, A total of 57 studies met the inclusion criteria, with data extracted from diverse populations across various continents, representing a comprehensive analysis of OSF prevalence globally. revise it.

4. The pooled prevalence of OSF was found to be higher in populations with a predominant habit of betel quid chewing, as compared to other regions where this habit is less common, indicating a strong correlation between the two factors. revise it.

>The literature review should include more critical discussion of the limitations of previous studies, specifically focusing on the geographical and methodological limitations.

>Some tables, particularly those summarizing the results of the meta-analysis, could benefit from additional footnotes or captions to clarify any complex data points. Ensure that all figures and tables are self-explanatory.

>A clearer explanation or documentation on how the raw data can be accessed and used, especially for those replicating the study.

>A more detailed discussion of the implications of the findings, particularly in how they address the stated hypotheses and contribute to the broader understanding of OSF.

Experimental design

>The manuscript could further emphasize the novelty of the research question by comparing it more explicitly with previous studies and highlighting the specific aspects that were lacking in earlier work.

>How data from different populations were handled and if any specific ethical guidelines were followed for the meta-analysis?

>Some sections of the methods could include the rationale behind selecting certain studies for inclusion in the meta-analysis and more detailed descriptions of the statistical models used.

Validity of the findings

>The manuscript does not explicitly focus on assessing the impact and novelty of the findings, which is appropriate for the context of validity review.

>The manuscript encourages replication, it could more clearly articulate the rationale for replication studies. Explaining the specific benefits to the literature from replicating this study would strengthen this aspect of the manuscript.

>Highlighting how replication could confirm or challenge the current findings would add depth to this section.

>The manuscript could provide more information on how potential confounding variables were controlled, especially in the meta-analysis.

>The authors avoid overgeneralizing their results, keeping the conclusions within the scope of the data collected and analyzed.

>The manuscript could enhance its discussion of the broader implications of the findings. Discussing how these results contribute to the wider field and suggest directions for future research would add value.

Additional comments

>The abstract is concise and informative, but consider adding a brief sentence on the implications of your findings. This would help readers quickly understand the significance of your study in the broader context of oral health research.

>Your discussion effectively interprets the results, it could benefit from a deeper exploration of the implications of your findings. Consider discussing how your results could influence future research, public health policies, or clinical practices, particularly in regions where OSF is most prevalent.

>A few additional citations from recent studies could be included to further substantiate your claims, particularly in the introduction where you discuss the epidemiology and risk factors of OSF.

>You have acknowledged certain limitations in your study, it might be helpful to elaborate on these in more detail. For instance, discussing the potential biases in the selected studies or the limitations of the meta-analytic approach could provide a more balanced view of your findings.

Reviewer 4 ·

Basic reporting

In the abstract, the PRISMA method can be added and has been registered with PROSPERO, for example This systematic review applied Preferred Reporting Items for Systematic Review and Meta-Analyses (PRISMA) 2020 etc.

Experimental design

It is well composed

Validity of the findings

Can information be added regarding how to diagnose OSF on study inclusion.
Can information be added criteria for habits and exposure, for example duration, frequency and intensity?

Additional comments

The systematic review is well composed.

---

## Round 0.2 · accepted · Accept

Dear authors, I consider your manuscript to be now ready for publication in PeerJ. Many congratulations.